# Aegis: Post-Training Attribute Unlearning in Federated Recommender Systems against Attribute Inference Attacks

## Abstract

As privacy concerns in recommender systems become increasingly prominent, federated recommender systems (FedRecs) have emerged as a promising distributed training paradigm. FedRecs enable the collaborative training of a shared global recommendation model without requiring the exchange of raw client interaction data. However, models trained using standard FedRec methods remain vulnerable to personal information leakage, particularly through attribute inference attacks, which can expose sensitive user attributes such as gender and race. In this paper, we address these user sensitive attributes as targets for federated unlearning. To protect users' sensitive information, attribute unlearning aims to eliminate sensitive attributes from user embeddings, thereby preventing inference attacks while preserving recommendation performance. We introduce a novel post-training federated unlearning framework, Aegis, which performs unlearning based on private attribute requests after the model has been trained, minimizing the degradation in recommendation accuracy. Aegis employs an information-theoretic multi-component loss function to balance privacy protection and recommendation performance. Additionally, Aegis adapts to scenarios where training interaction data may be unavailable, reflecting real-world centralized protection scenarios. Comprehensive evaluations on various benchmark datasets demonstrate that our proposed method effectively safeguards user privacy while maintaining high-quality recommendations.

## Keywords

Federated Learning, Recommender System, Attribute Unlearning, Privacy-preserving.

## 1 INTRODUCTION

Recommender systems (RS) have become fundamental to modern web applications, driving personalized user experiences across platforms like e-commerce [5, 33] and streaming services [31, 32] by utilizing vast amounts of user interaction data. However, traditional RS often require users' personal data—such as browsing history and purchasing behavior—to be centralized on remote servers for processing, which raises significant privacy risks including the potential data breaches. To address these concerns, federated recommender systems (FedRecs) [2, 47, 48] have emerged as a promising solution that aggregates model updates on the server while keeping user interaction data local and performing model training on the client side. Despite this improvement, a critical challenge persists with stricter privacy requirements: recent privacy regulations such as the GDPR [57] and CCPA [26] emphasizing the *"right to be forgotten"* has promoted users' demand for the forgetting of personal private attributes in recommender systems, i.e. *attribute unlearning* [22]. This concept pertains to the removal of the inherent attributes of user embeddings in the trained model, such as race and gender, which are not used as training targets.

Attribute unlearning is intuitively important for privacy protection in recommender systems. Although existing FedRecs avoid the transmission of raw data, model training based on historical user behavior may still be vulnerable to *attribute inference attacks (AIAs)* [4, 30], which potentially reveal sensitive user attributes [20, 40]. Research [60] has demonstrated that basic machine learning models can successfully infer user attributes from user embeddings learned by collaborative filtering models. We conducted attribute inference attacks on FedNCF [51] systems trained on dataset MovieLens-100K (ML-100K) and MovieLens-1M (ML-1M) [23]. As shown in Table 1, the accuracy of attribute inference attackers is consistently higher than that of random attackers, revealing significant privacy leakage risks.

Some existing machine unlearning methods [6, 64] aim to make the unlearned model as consistent as possible with one retrained from scratch. However, in the context of attribute unlearning, this fails to decouple latent attributes from the model, hindering effective unlearning. Unlike regular machine forgetting, attribute unlearning cannot simply erase specific attribute traces. Moreover, most current approaches to attribute privacy protection in recommender systems operate during training, relying on network modifications [22, 25, 68, 69] or adversarial training [3, 18, 20, 38]. These in-training attribute-preserving methods are costly, complex, and require prior knowledge of privacy issues, making them less suited to dynamic privacy needs. In real-world scenarios, users' privacy requirements may change over time, and federated clients may want to adjust their privacy settings after training rather than determining them beforehand. This calls for a post-training federated attribute unlearning method that can handle dynamic requests without needing full retraining or redesign of the existing model structure. Furthermore, attribute unlearning requests are often unpredictable, and training data or historical updates may be inaccessible due to privacy regulations or data deletion [11]. Thus, federated attribute unlearning must work both with and without access to the interaction training data. In conclusion, a flexible and efficient post-training framework is essential to address evolving privacy demands in federated recommender system environments.

Existing methods protect privacy by artificially designed adding noise to user embeddings, such as through local differential privacy [1], but this often degrades recommendation performance [40]. Additionally, ensuring unlearning effectiveness is challenging, as attacks may come from complex machine learning or deep learning models [44], whose mechanisms are not fully understood [29]. In this paper, we focus on protecting trained user embeddings from potential attacks with two key objectives: i). making private attributes indistinguishable in the embeddings to reduce the success of inference attacks, and ii). preserving recommendation performance, as both users and service providers seek to avoid significant decline in quality. These objectives guide the design of our system.

**Table 1: FedRec Recommendation Utility and Attribute Inference Attack Results on Different Datasets**

| Dataset | Utility | | Attribute Privacy | | |
|---------|---------|--------|--------|--------|------------|
| | NDCG@10 | HR@10 | Gender | Age | Occupation |
| **ML-100K** | 0.708 | 0.680 | 0.714 | 0.280 | 0.149 |
| **ML-1M** | 0.699 | 0.684 | 0.849 | 0.353 | 0.119 |
| **Random Attacker** | | | 0.500 | 0.143 | 0.048 |

To achieve these objectives, we propose the Aegis[1] framework, an innovative approach that seamlessly integrates attribute unlearning with performance retention. Our method fine-tunes a pre-trained recommender system to safeguard sensitive user attributes from attribute inference attacks. It employs a multi-component loss function grounded in information theory [54] to address the optimization problem. Specifically, it reduces the association between user embeddings and sensitive attributes while maintaining recommendation performance. Additionally, it includes a regularization component to ensure stability in user embeddings. Our framework not only facilitates localized training for attribute unlearning but also adapts to scenarios where client interaction data is inaccessible through centralized unlearning methods. We summarize our main contributions as follows:

- To the best of our knowledge, we are the first to investigate the post-training attribute unlearning in federated recommender systems, addressing scenarios both with and without access to training interaction data. This approach reflects more realistic privacy-preserving measures.
- We propose the Aegis framework for federated attribute unlearning, formalizing the setting of attribute attacks and identifying two key objectives: attribute indistinguishability and recommendation performance retention. Aegis leverages information-theoretic principles by introducing a multi-component loss function that synchronously optimizes both objectives, balancing privacy and recommendation accuracy.
- We implemented the Aegis system and conducted extensive evaluation on benchmark datasets to evaluate our method's performance in terms of attribute unlearning and recommendation knowledge retention. Results demonstrate that our framework effectively balances privacy and performance.

## 2 PRELIMINARIES AND OBJECTIVES

### 2.1 Federated Recommendation

We first describe the general mathematical formulation of a federated recommender system. Let $U$ represent the set of users with the total number of users denoted as $|U|$, and $V$ represent the set of items with the total number of items denoted as $|V|$. Users across different clients collectively form the $U$. Each user $u_i \in U$ $(1 \le i \le |U|)$ owns a local dataset $D_i$, which is defined as: $D_i = \{(u_i, v_j, r_{ij})|v_j \in V\}$, where $r_{ij} = 1$ indicates that user $u_i$ has interacted with item $v_j$, and $r_{ij} = 0$ means no interaction, in which case $v_j$ is considered a negative sample. The goal of the federated recommender system is to predict the score $\hat{r}_{ij} = s_\psi(\mathbf{em}_i, \mathbf{em_j})$ of user $u_i$ on non-interacted items $v_j$, thereby generating a recommendation list $\hat{V}_i$, satisfying:

$$\hat{V}_i = \text{Top-K}(\{\hat{r}_{ij}|v_j \in V \setminus D_i\}), \quad (1)$$

---

[1]From Greek mythology, a powerful shield used by Zeus or Athena to fend off attacks.

where $s_\psi(\cdot)$ is a score function, which can be a dot product, a multi-layer perceptron, etc. $\mathbf{em}_i = f_p(f_\varphi(u)) \in \mathbb{R}^d$ and $\mathbf{em}_j = f_p(f_\varphi(j)) \in \mathbb{R}^d$ represent the embeddings of users and items, where $d$ is the embedding dimension. The function $f_\varphi$ represents an embedding layer that maps users/items to vectors, and $f_p$ represents a propagation layer that captures collaborative signals [63]. FedRecs train the model across multiple distributed clients, such as users' mobile devices or computers. The central server does not directly access users' interaction data but instead coordinates multiple rounds of local training on each client and aggregates the model parameter updates uploaded by each client to form a global model. Specifically, in each global training round $t$: First, the central server distributes the global model parameters to each selected user/client $i$, $\theta_{i,t}^0 = \theta_{g,t}$. Each client combines the received global parameters with their local user embeddings to form a local recommender model. Then, the local recommender is optimized using the local dataset as below:

$$\theta_{i,t}^{t_l+1} = \theta_{i,t}^{t_l} - \eta \nabla_{\theta_i} \mathcal{L}(\theta_{i,t}, D_i), \quad (2)$$

where $\eta$ is the learning rate, and $\mathcal{L}$ is the loss function (e.g., BPR loss [52]). After $T_l$ rounds of local training, each client sends the updated global parameters $\theta_{i,t}^{T_l}$ (or the parameter updates $\Delta\theta_{i,t}^{T_l} = \theta_{i,t}^{T_l} - \theta_{i,t}^0$) back to the server. The server aggregates the received parameter using a specific aggregation strategy [46], such as:

$$\theta_{g,t+1} = \theta_{g,t} + \frac{1}{|U|}\sum_{i=1}^{|U|}\Delta\theta_{i,t}^{T_l} \quad \text{or} \quad \theta_{g,t+1} = \frac{1}{|U|}\sum_{i=1}^{|U|}\theta_{i,t}^{T_l}. \quad (3)$$

The above steps are repeated until the convergence condition is met, i.e., the performance of the model reaches the predefined standard.

### 2.2 Attribute Inference Attack

In recommender systems, attackers can infer private user attributes (such as gender, age, or race) based on the embedding data between users and items, as well as the trained model [20, 40]. This leads to attribute inference attacks (AIA) [4, 30, 68], which pose a significant threat to user privacy. A key issue in AIA is the unintentional leakage of information about non-target attributes during the recommendation process. To mitigate this risk, the concept of attribute unlearning has been introduced, which allows a system to "forget" certain sensitive attributes post-training while maintaining recommendation performance.

**Threat Model**: The threat model is illustrated in Figure 1: in federated learning (FL) environments, malicious servers may infer users' sensitive attributes by accessing user embedding vectors from FL clients, leading to privacy breaches. In this scenario, we assume the attacker adopts a grey-box attack strategy, meaning the attacker cannot access all model parameters but can access some user embedding vectors $\mathbf{em}_i$ and corresponding attribute information $z_i$. The threat model's goal is to infer the private attribute $z_i$ from the user embedding $\mathbf{em}_i$. In this paper, this attack is framed as a classification problem, where the attacker employs a classification model $g$ to predict the private attributes.

During the training phase of the threat model, we assume that the attacker does not have direct access to the original dataset. Instead, the attacker uses a shadow dataset $D_{\text{shadow}}$ to train the model. This shadow dataset can be generated by sampling from the original user data or from other users within the same distribution [40, 53]. The

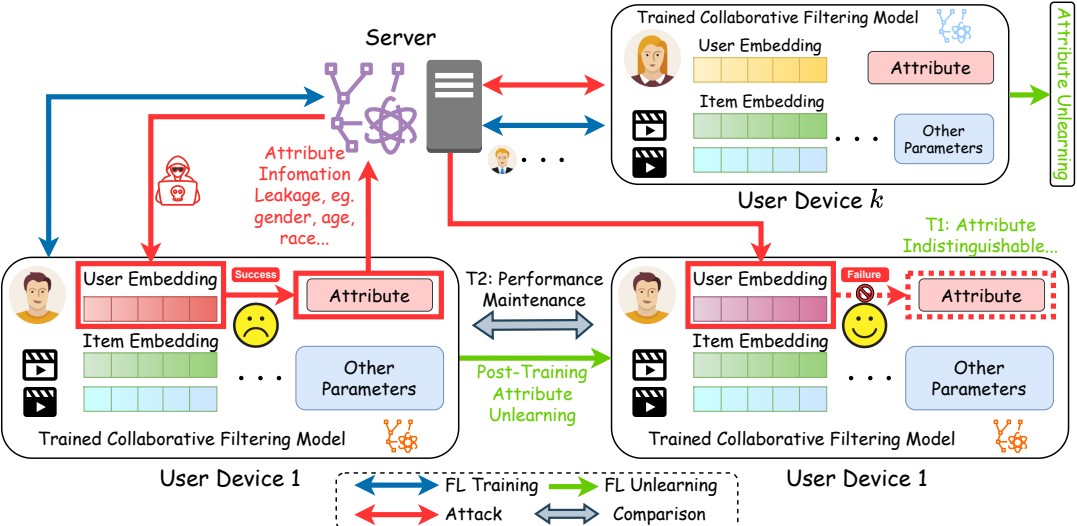

**Figure 1: Overview of Post-Training Attribute Unlearning in Federated Recommender Systems.**

input to the shadow dataset consists of user embeddings $\mathbf{Em}_{\text{shadow}}$, while the target comprises the attribute labels $Z_{\text{shadow}}$. Although using a shadow dataset may reduce the overall effectiveness of the attack, this assumption is reasonable, as assuming access to the full dataset would be overly idealistic and impractical. During training, the attacker constructs the threat model $g$ by minimizing the classification loss on the shadow dataset:

$$\min_{g} \mathbb{E}_{(\mathbf{Em}_{\text{shadow}}, Z_{\text{shadow}})} \left[ \mathcal{L}(Z_{\text{shadow}}, g(\mathbf{Em}_{\text{shadow}})) \right]. \tag{4}$$

In the inference phase, the attacker uses the trained model $g$ to predict the attribute for a new user embedding $\mathbf{em}_i$. This results are in the prediction $\hat{Z}_i = g(\mathbf{Em}_i)$, where $\hat{Z}_i$ represents the attacker's estimated value of the private attributes of the user set $\mathbf{Em}_i$.

## 2.3 Our Objectives

In FedRecs, we collaboratively train a collaborative filtering model using data distributed across different clients in a privacy-preserving manner. Notice that users have dynamic privacy preferences, with the private attribute set $A_u$. Given a trained global model, we aim to perform "unlearning" on the private attribute set $A_u$ for all the users $u_i \in U$, i.e., to generate new user embeddings $\mathbf{em}'_i$ from the original user embeddings $\mathbf{em}_i$ to mask the private attribute. To achieve this, we need to strike an optimal balance between privacy protection and recommendation performance. We formalize federated attribute unlearning as an optimization problem as follows:

# Objective 1: Unlearning Objective

Ensure that the user attribute $au_i$ cannot be easily distinguished from different user embeddings, thereby protecting the privacy of the attribute information from potential attackers. We define a function $dg$ to evaluate the distinguishability of the attributes:

$$\min_{\mathbf{em}'_i} \sum_{au_j \in A_u} D(dg(\mathbf{em}'_i), au_j), \tag{5}$$

where $D$ is a measure of attribute distinguishability.

# Objective 2: Recommendation Objective

Ensure the performance of the FedRecs is maintained to avoid impacting the original recommendation quality. We can measure the change in recommendation performance as below:

$$\min_{\mathbf{em}'_i} \quad \text{Dist}(M(\mathbf{em}_i), M(\mathbf{em}'_i)), \tag{6}$$

where Dist is a measure of the change in performance.

Additionally, to ensure the efficiency of the unlearning process, we need to limit the time overhead of unlearning. Combining the above objectives, we provide a systematic approach to achieve "unlearning" of user attributes while ensuring a high balance between privacy protection and recommendation performance.

## 3 POST-TRAINING FEDERATED ATTRIBUTE UNLEARNING FOR RECOMMENDATION

In this section, we demonstrate how Aegis achieve attribute indistinguishability through model fine-tuning with a carefully designed multi-component federated loss function.

## 3.1 Overview

Our Aegis framework is a complement to common FedRecs and is applicable to systems based on different training methods. Given the dynamic private attribute needs of federation clients, Aegis fine-tunes a trained federated recommender system to protect sensitive attributes from attribute inference attacks.

Aegis advocates two key properties for effective attribute unlearning: i) **Private Attribute Unlearning**, which effectively removes the association between user-marked attributes for deletion and user embeddings to prevent privacy leakage; and ii) **Recommendation Knowledge Retention**, which ensures that recommendation performance is maintained post-unlearning. Aegis follows a client-server architecture in federated learning, where clients update user embeddings based on unlearning methods using stored interaction data to eliminate the relationship between sensitive attributes and embeddings. The updated embeddings and other

model parameters are then uploaded to the server, where FedAvg gradient aggregation [46] is performed, and the updated global model is sent back to the clients for iterative optimization. When post-training attribute unlearning is required but client training data is unavailable, Aegis also allows centralized unlearning to be executed solely on the client side.

## 3.2 Compositional Attribute Unlearning

To achieve attribute unlearning in privacy-sensitive scenarios, we proposes an information theory [54] based multi-component loss function, treating it as an optimization problem for user embeddings. The core idea is to identify the mutual information between the embedding distribution and the attributes, effectively capturing their relationship, and then applying forgetting based on the privacy requirements. Our approach involves designing an appropriate loss function and using optimization techniques to obtain the desired user embeddings, temporarily disregarding intermediate operations and transformations. The goal is to ensure that the model's embedding $\mathbf{em}'$ achieves the following: i) Discard information related to specific (e.g., sensitive or private) attributes. ii) Retain information relevant to the recommendation task. This objective can be formalized as:

- Minimize the mutual information between the user embedding $\mathbf{em}'_i$ and the attributes to be forgotten $au_j \in A_u$, defined as:

$$\min_{\mathbf{em}'_i} \sum_{au_j \in A_u} \mathcal{I}(\mathbf{em}'_i; au_j); \tag{7}$$

- Maximize the mutual information between the embedding $\mathbf{em}'_i$ of user $u_i$ and the primary recommendation task, defined as:

$$\max_{\mathbf{em}'_i} \mathcal{I}(\mathbf{em}'_i, V_{EM}; R_i); \tag{8}$$

where $V_{EM}$ represents the item embeddings of items $V$ and $R_i = [r_{i,j}]_{j=1}^{|V|}$ represents the interaction matrix between $u_i$ and $V$.

We propose a multi-component loss function, with each component specifically designed to address one of the above goals.

### 3.2.1 Private Attributes Information Loss. Directly computing mutual information $\mathcal{I}(\mathbf{em}'_i; au_j)$, $au_j \in A_u$ is challenging because it requires estimating joint and marginal probability distributions, a process that becomes computationally expensive in high-dimensional embedding spaces. This difficulty arises due to unknown real distributions, complex non-linear dependencies, and the curse of dimensionality. Therefore, exact estimation of mutual information is impractical without the use of approximation techniques. To address this, we approximate the mutual information using a variational upper bound based on the Kullback–Leibler (KL) divergence [36], which measures the difference between two probability distributions. The resulting loss function is as below:

$$\mathcal{L}_{i,j}^{\mathrm{AU}} = I(\mathbf{em}_i; au_j)$$
$$\approx \frac{1}{|U_{au_j}|} \sum_{k=1}^{|U_{au_j}|} D_{\mathrm{KL}}(q_\phi(\mathbf{em}_i|X_{au_j=C_k}) \| p(\mathbf{em}_i|X)), \tag{9}$$

where $|U_{au_j}|$ represents the number of different labels in attribute $au_j$, and $X_{au_j=C_k}$ represents the input $X$ where the label of attribute $au_j$ is $C_k$. $D_{\mathrm{KL}}(q\|p)$ denotes the Kullback–Leibler divergence between the variational distribution $q_\phi(\mathbf{em}_i|X_{au_j=C_k})$, which approximates the embedding distribution conditioned on the attribute

$au_j = C_k$, and the prior distribution $p(\mathbf{em}_i|X)$, which represents the embedding distribution conditioned only on the input $X$. By minimizing this KL divergence, we effectively reduce the information in the embedding $\mathbf{em}_i$ that is related to the attribute $au_j$, thus achieving the goal of unlearning the attribute. Intuitively, we aim to compute the distribution of different attribute classes $au_j$ and minimize their KL divergence from the same global distribution, ensuring that the embedding data of users across different classes cannot be identified. To efficiently compute the distribution of user embeddings for the $|U_{au_j}|$ distinct classes within $au_j$ and subsequently calculate the KL divergence, we perform the following operations on each client:

First, we compute the user embedding distribution for each class. Since Aegis is a post-training method, the user embedding data is already available before the unlearning process begins, with each class having an associated set of user embeddings. The probability distribution of embeddings for each class can be estimated, for instance, by fitting a Gaussian distribution or another suitable model [14, 17, 24, 59]. In this work, as our user embeddings are represented in a continuous vector space, we fit a Gaussian distribution to each class's embedding distribution. For each attribute class $C_k \in au_j$, the mean vector $\mu_{j,k}$ is computed as below:

$$\mu_{j,k} = \frac{1}{|S_{au_j=C_k}|} \sum_{\mathbf{em}_i \in S_{au_j=C_k}} \mathbf{em}_i, \tag{10}$$

where $S_{au_j=C_k} = \{\mathbf{em}_i \mid au_j(\mathbf{em}_i) = C_k\}$ represents the set of user embeddings where the attribute $au_j$ belongs to class $C_k$, and $|S_{au_j=C_k}|$ denotes the number of embeddings in this set. We compute the covariance matrix $\Sigma_i$ of the Gaussian distribution as below:

$$\Sigma_{j,k} = \frac{1}{|S_{au_j=C_k}|} \sum_{\mathbf{em}_i \in S_{au_j=C_k}} (\mathbf{em}_i - \mu_{j,k})(\mathbf{em}_i - \mu_{j,k})^T. \tag{11}$$

Second, we compute the user embedding distribution for the global set of embeddings by aggregating all the embeddings across classes. The global mean vector $\mu_{\mathrm{global}}$ is computed as:

$$\mu_{\mathrm{global}} = \frac{|S_{au_j=C_k}|}{N} \sum_{k=1}^{N} \mu_{j,k}, \tag{12}$$

where $N$ is the total number of user embeddings across all classes. Similarly, the global covariance matrix $\Sigma_{\mathrm{global}}$ is computed as:

$$\Sigma_{\mathrm{global}} = \frac{1}{N} \sum_{i=1}^{N} (\mathbf{em}_i - \mu_{\mathrm{global}})(\mathbf{em}_i - \mu_{\mathrm{global}})^T. \tag{13}$$

This global Gaussian distribution captures the overall structure of the user embeddings across all attribute classes.

Finally, we calculate the KL divergence between each class embedding distribution and the global embedding distribution. Notably, there is an analytical solution for computing the KL divergence between two multivariate Gaussian distributions. For a class $C_k$ and the global Gaussian distributions $\mathcal{N}(\mu_{j,k}, \Sigma_{j,k})$ and $\mathcal{N}(\mu_{\mathrm{global}}, \Sigma_{\mathrm{global}})$, the formula for the KL divergence is:

$$D_{\mathrm{KL}}(\mathcal{N}(\mu_{j,k}, \Sigma_{j,k}) \| \mathcal{N}(\mu_{\mathrm{global}}, \Sigma_{\mathrm{global}})) = \frac{1}{2} \left( \log \frac{\det(\Sigma_{\mathrm{global}})}{\det(\Sigma_{j,k})} - d \right)$$
$$+ \frac{1}{2} \left( \mathrm{Tr}(\Sigma_{\mathrm{global}}^{-1} \Sigma_{j,k}) + (\mu_{\mathrm{global}} - \mu_{j,k})^T \Sigma_{\mathrm{global}}^{-1}(\mu_{\mathrm{global}} - \mu_{j,k}) \right), \tag{14}$$

where $d$ is the dimensionality of the user embeddings, $\det(\Sigma)$ is the determinant of the covariance matrix, and Tr denotes the trace operation. By fitting a Gaussian distribution to the embeddings, we compute the mean and covariance matrix for each class, and

then use the KL divergence to minimize the difference between the user embedding distributions of different classes. This allows us to calculate the unlearning loss function $\mathcal{L}_j^{AU}$ for $au_j$ as below:

$$\mathcal{L}_j^{AU} = \frac{1}{|U_{au_j}|} \sum_{k=1}^{|U_{au_j}|} D_{\text{KL}}(\mathcal{N}(\mu_{j,k}, \Sigma_{j,k}) \| \mathcal{N}(\mu_{\text{global}}, \Sigma_{\text{global}})). \quad (15)$$

Minimizing the loss function $L_j^{AU}$ enables the unlearning of the private attribute $au_j$, thereby providing protection against AIAs.

### 3.2.2 *Recommendation Knowledge Retention Loss*.
Since performing unlearning may lead to a degradation in recommendation performance, additional design is necessary to achieve objective 2. An intuitive approach is to directly use the recommendation loss function from the federated training phase (e.g., binary cross-entropy (BCE) [65], root mean squared error (RMSE) [27], or Bayesian personalized ranking (BPR) [52] loss) as the optimization objective to maintain recommendation quality. This recommendation loss $\mathcal{L}^{Rec}$ can be defined as:

$$\mathcal{L}^{Rec} = \mathcal{L}_{BCE/BPR,\dots}(s_\psi(f_{\varphi,p}(u), f_{\varphi,p}(i)), \mathbf{R}), \quad (16)$$

where $\mathbf{R}$ is the interaction matrix, and each element $r_{i,j} \in \mathbf{R}$ represents the interaction between user $u_i$ and item $v_j$.

To accelerate the execution process, we only update user embeddings during unlearning, so we additionally propose the use of a regularization loss $\mathcal{L}^{Reg}$ to restrict the range of user embedding updates, preventing drastic changes in user embeddings and thus leveraging the prior learning. The L2-regularization [21, 40] term is defined as below:

$$\mathcal{L}^{Reg} = \sum_{i=1}^{|U|} \|\mathbf{em}_i - \mathbf{em}_i'\|_2^2 = \sum_{i=1}^{|U|} \sum_{j=1}^{d} (em_{i,j} - em_{i,j}')^2, \quad (17)$$

where $\mathbf{em}_i$ and $\mathbf{em}_i'$ represent the user embeddings before and after unlearning, respectively. Since the interaction data may be inaccessible due to privacy restrictions or data modifications after training, the loss $\mathcal{L}^{Rec}$ might no longer be applicable. In such scenarios, the regularization term $\mathcal{L}^{Reg}$ will help preserve recommendation performance. The underlying rationale is that closer model parameters typically lead to more consistent model performance.

### 3.2.3 *Summary*.
Eq. (15), (16), and (17) in Aegis represent two sub-objectives corresponding to the motivations: Eq. (15) focuses on the elimination of sensitive attribute information, and Eq. (16) and Eq. (17) aim to enhance recommendation performance. For each user, we achieve private user embedding training for each client through a federated training process. In each unlearning round, we sample a set of users and their historical interaction data from $U$. By jointly learning $\mathcal{L}^{AU}$, $\mathcal{L}^{Rec}$, and $\mathcal{L}^{Reg}$, the private embedding unlearning objective for clients can be formulated as:

$$\varphi, p = \arg\min_{\varphi,p} \mathcal{L}^{All} = \arg\min_{\varphi,p} \mathcal{L}^{Rec} + \beta \mathcal{L}^{Reg} + \gamma \sum_{j \in A_u} \mathcal{L}_j^{AU}, \quad (18)$$

where $\beta$, and $\gamma$ are hyperparameters that balance the trade-off between recommendation utility and privacy protection. A larger $\gamma$ indicates stronger protection of attribute privacy, while larger $\beta$ enhance recommendation accuracy. It is worth noting that the weight for the privacy loss of each attribute can be adjusted; for simplicity, we assume users weigh all private attributes equally. When the training interaction data is unavailable and the recommendation loss cannot be applied, we only use the regularization term to prevent excessive degradation of recommendation performance. The

---

**Algorithm 1** Aegis Attribute Unlearning Process

---

**Input:** Trained model parameters $\theta_0 = \{\varphi_0, p_0\}$, $\psi_0$, private attributes $A_u$, number of local iterations $L_l$, server aggregation rounds $T$, Hyperparameters $\beta, \gamma$, threshold $\epsilon$, learning rate $\eta$.

**Output:** Updated user embeddings $f_\theta(u)$ and model parameters $\theta$

1: The client submits an unlearning request for the attribute set $A_u$;
2: Initialize global parameter with trained model $\theta_0 = \{\varphi_0, p_0\}, \psi_0$;
3: **Server:** Distribute the attribute unlearning request of $A_u$ to clients;
4: **if** The clients can participate in unlearning with available data **then**
5:    **for** each global round $t = 0, 1 \dots, T-1$ **do**
6:       sampling a fraction of clients $C_t \subseteq U$
7:       **for** each client $c_i \in C_t$ in parallel **do**
8:          **for** each local iteration $l = 1, 2, \dots, L_l$ **do**
9:             Compute the private attribute unlearning loss $\mathcal{L}_t^{AU} = \sum_{j \in A_u} \mathcal{L}_{j,t}^{AU}$ using Eq. (15)
10:            Compute the recommendation knowledge retention loss $\mathcal{L}_t^{Rec}$ and regularization loss $\mathcal{L}_t^{Reg}$ using Eq. (16) and (17)
11:            Update model parameters $\theta_{t,i} = \{\varphi_{t,i}, p_{t,i}\}$ by minimizing $\mathcal{L}^{All}$ in Eq. (18): $\theta_{t,i} = \theta_{t,i} - \eta \nabla_\theta (\mathcal{L}_t^{Rec} + \beta \mathcal{L}_t^{Reg} + \gamma \mathcal{L}_t^{AU})$
12:          **end for**
13:          Upload updated unlearning gradients to server
14:       **end for**
15:       **Server:** Aggregate the gradients and update the global model, $\theta_{t+1} = \frac{1}{\sum_{c_j \in C_t} |D_{c_j}|} \sum_{c_i \in C_t} |D_{c_i}| \theta_{t,i}$
16:       **if** The update in embeddings: $\|f_{\theta_{t+1}}(u) - f_{\theta_t}(u)\| < \epsilon$ **then**
17:          **Break:** End training early as updates are below the threshold
18:       **end if**
19:       **Server:** Distribute $\theta_{t+1}$ to all clients, $\theta_{t+1,i} = \theta_{t+1}, \forall i \in U$
20:    **end for**
21: **else**
22:    **for** each round $t = 0, 1, \dots, T-1$ on **Client making request do**
23:       Compute the private attribute unlearning loss $\mathcal{L}_t^{AU} = \sum_{j \in A_u} \mathcal{L}_{j,t}^{AU}$ using Eq. (15)
24:       Compute the regularization loss $\mathcal{L}_t^{Reg}$ using Eq. (17)
25:       Update model parameters $\theta = \{\varphi, p\}$ by minimizing $\mathcal{L}^{All'}$ in Eq. (19): $\theta_{t+1} = \theta_t - \eta \nabla_\theta (\beta \mathcal{L}_t^{Reg} + \gamma \mathcal{L}_t^{AU})$
26:       **if** The update in embeddings: $\|f_{\theta_{t+1}}(u) - f_{\theta_t}(u)\| < \epsilon$ **then**
27:          **Break:** End training early as updates are below the threshold
28:       **end if**
29:    **end for**
30: **end if**

---

training objective in this data-free case is:

$$\varphi, p = \arg\min_{\varphi,p} \mathcal{L}^{All'} = \arg\min_{\varphi,p} \beta \mathcal{L}^{Reg} + \gamma \sum_{j \in A_u} \mathcal{L}_j^{AU}. \quad (19)$$

Finally, we use stochastic gradient descent (SGD) [34] on each client to optimize the total loss function $\mathcal{L}^{All}$ or $\mathcal{L}^{All'}$.

## 3.3 Training Process

Aegis operates after federated learning, focusing on fine-tuning to ensure privacy by unlearning specific attributes. Two models are involved in our framework: the embedding network for recommendation $f_{p,\varphi}$ and the recommendation score function $s_\psi$.

### 3.3.1 *Federated Learning (Pre-Unlearning Stage)*.
We adopt the standard FedRec model for training the recommender system. In each training round, the central server begins by sampling a group of users and distributing the model parameters. The clients then perform local training, iterating over mini-batches to compute the

loss and update the weights of $f_{p,\varphi}$ and $s_\psi$. After $L_l$ local iterations, the clients upload the weights to the server. Finally, the server aggregates the weights using FedAvg [46] and updates the model.

*3.3.2* ***Attribute Unlearning Fine-tuning***. As users' privacy preferences may change over time, when attributes that were previously considered non-sensitive become sensitive, Aegis needs to promptly provide protection. Based on the set of private attributes that need protection, we perform post-training fine-tuning of the trained recommender system, as shown in Algorithm 1. Aegis operates in two protection modes as below:

**i) Aegis-Fed (federated protection):** When clients are willing to participate in the fine-tuning process and provide the relevant recommendation data for unlearning, we use $\mathcal{L}^{\text{All}}$ from Eq. (18) as the local loss function and perform local training updates. The server handles aggregation and model distribution, following the usual federated training process.

**ii) Aegis-CS (centralized protection):** When clients are unwilling to participate in the fine-tuning process or are unable to provide the necessary recommendation data due to privacy concerns or data changes, we use $\mathcal{L}^{All'}$ from Eq. (19) as the loss function to conduct fine-tuning on the client making the unlearning request. We iteratively update the user embeddings until the update difference is less than a threshold $\epsilon$, i.e., $\|e_{u,k+1} - e_{u,k}\| \le \epsilon$.

## 4 Evaluation

We evaluated our Aegis method on four benchmark datasets, demonstrating its performance in attribute unlearning.

### 4.1 Evaluation Setup

**Testbed.** We implemented Aegis using Python 3.8.0 and PyTorch 2.2.0, and run all experiments on NVIDIA A100 Tensor Core GPUs.
**Datasets.** The experiments were conducted on four publicly accessible datasets as detailed in Table 2, each representative of various web applications. These include the movie rating datasets MovieLens-100K and MovieLens-1M [23] used in media streaming platforms, the clothing sales dataset ModCloth [58] relevant to e-commerce applications, and the music listening behavior dataset [7] employed in music recommendation services. The datasets encompass user-item interactions as input data, along with user attributes such as gender and age, making them well-suited for RS research. The private attribute 'Age' is divided into seven age groups following the method used in the MovieLens-1M. For location labels, we use continent tags based on the countries from the Last.FM-1K dataset.
**FedRec Models and Hyperparameters.** We use FedNCF [51] as our foundational FedRecs and employ dot product as the scoring function. We set the dimensions of user and item embeddings to 128 and use SGD [34] as the optimization algorithm with a learning rate of 0.01. For the base training loss function $\mathcal{L}^{Rec}$, we use BPR loss [53] to train the recommendation model. We use 10 federated clients, with default hyperparameters set to $\beta = 0.1$, and $\gamma = 10$.
**Attacker Seeting.** For selecting the attribute inference model for user embedding attacker, we utilize easily implementable and powerful machine learning models, including a three-layer MLP model [49] and the XGBoost model [16]. Both models are employed as private attribute classifiers and trained on shadow datasets.

**Table 2: Summary of Datasets**

| Dataset | Users | Items | Ratings | Density |
|---|---|---|---|---|
| MovieLens-100K | 943 | 1,682 | 100,000 | 6.30% |
| MovieLens-1M | 6,040 | 3,952 | 1,000,209 | 4.19% |
| ModCloth | 44,784 | 1,020 | 99,893 | 0.22% |
| Last.FM-1K | 992 | 176,948 | 19,150,868 | 10.91% |

**Evaluation Metrics.** In evaluating recommendation performance, we employ metrics widely used in recommender systems, reporting recommendation utility by calculating the average hit ratio (HR) [12] and normalized discounted cumulative gain (NDCG) [28] across the ranked item lists of all test users. We truncate the ranked lists for both metrics at positions 5 and 10. For privacy-preserving performance evaluation, we assess information leakage in user embeddings using the accuracy of attribute classifiers. The AIA's goal is high attack accuracy, but excessively low accuracy could trigger the "Streisand Effect" [10, 11, 19], inadvertently leading to privacy exposure. Our goal is to protect against AIAs, where scores closer to those of a random attacker indicate better privacy preservation.
**Unlearning Methods.** There are numerous studies on federated unlearning, but most of them are not applicable to the attribute unlearning problem. To the best of our knowledge, we are the first to study post-training federated attribute unlearning. We introduce two versions of Aegis: Aegis-Fed and Aegis-CS. The key difference lies in whether client interaction training data is involved during the unlearning process, corresponding to data-dependent **(DD)** and data-free **(DF)** settings, respectively. We compare our methods with existing defenses against attribute inference attacks:

- **UC-FedRec [25]**: A federated AIA defense method performed during training. It modifies the original federated recommender system by training attribute filters for each client, minimizing the attribute classification loss during training to achieve attribute unlearning. Although the background setting differs from our post-training approach, the comparison aids in a comprehensive understanding of the attribute unlearning problem.
- **U2U-R** and **D2D-R [40]**: The post-training attribute unlearning methods exclude training data, using user-to-user (U2U) loss and distribution-to-distribution (D2D) distance loss as attribute distinguishability losses to achieve unlearning, respectively. We extend it to the federated setting with multiple attribute labels.

### 4.2 Results and Analysis

*4.2.1* ***Attribute Unlearning Performance***. The classification accuracy of attackers across different datasets reflects Aegis's performance of attribute unlearning, with results shown in Table 3. We treat the gender, age groups, and occupation of users in MovieLens-100K and MovieLens-1M, the body shape of users in ModCloth, and the gender, age groups, and country location of users in Last.FM-1K as sensitive attributes. We can draw the following conclusions: First, for FedRec without any protective measures, the XGBoost attack on the original user embedding achieves an average improvement of 21.54%, and the MLP attack improves by 21.15% compared to a random attacker. This indicates that private information in user embeddings can be leaked to attackers. Second, in the data-dependent DD scenario, for MLP attackers, both Aegis-Fed and UC-FedRec methods reduce MLP attack performance, with a reduction of 14.53% and 14.63%, respectively. For XGBoost attackers,

**Table 3: Results of Unlearning Performance (Attack Accuracy of XGBoost/MLP Attackers), where DD Indicates the Unlearning Process is Dependent on the Training Interaction Data, and DF Indicates that the Unlearning Process is Interaction Data-free.**

| Dataset | | | MovieLens-100K | | | MovieLens-1M | | | ModCloth | Last.FM-1K | | |
|---|---|---|---|---|---|---|---|---|---|---|---|---|
| Sensitive Attributes | | | Gender | Age | Occupation | Gender | Age | Occupation | Body Shape | Gender | Age | Location |
| **XGBoost Attacker** | | Original | 0.7143 | 0.2804 | 0.1490 | 0.8487 | 0.3526 | 0.1192 | 0.7419 | 0.5989 | 0.4828 | 0.5604 |
| | DD | Aegis-Fed | **0.5703** | **0.2222** | **0.1058** | **0.5968** | **0.1798** | **0.0589** | **0.5316** | **0.4798** | **0.2778** | **0.3586** |
| | | UC-FedRec | 0.6772 | 0.2857 | 0.2011 | 0.7280 | 0.3220 | 0.1543 | 0.7325 | 0.5606 | 0.2980 | 0.4899 |
| | DF | Aegis-CS | **0.5450** | **0.2116** | **0.0794** | **0.6450** | **0.2064** | **0.0766** | **0.5709** | 0.4506 | **0.2929** | **0.3766** |
| | | U2U-R | 0.9921 | 0.5947 | 0.6980 | 0.9997 | 0.9998 | 0.9940 | 0.9999 | 0.9999 | 0.8690 | 0.9518 |
| | | D2D-R | 0.5834 | 0.2464 | 0.1020 | 0.6821 | 0.2941 | 0.0923 | 0.7799 | **0.4746** | 0.4585 | 0.4876 |
| **MLP Attacker** | | Original | 0.7105 | 0.3386 | 0.0954 | 0.7310 | 0.3470 | 0.1159 | 0.7654 | 0.6102 | 0.5350 | 0.5604 |
| | DD | Aegis-Fed | 0.6541 | **0.1376** | 0.0794 | **0.5625** | **0.1598** | 0.0762 | **0.6602** | **0.5556** | **0.1162** | 0.2828 |
| | | UC-FedRec | **0.6085** | 0.1640 | **0.0582** | 0.3654 | 0.1836 | **0.0263** | 0.6784 | 0.4242 | 0.1364 | **0.2171** |
| | DF | Aegis-CS | **0.6224** | **0.1693** | 0.0688 | **0.6032** | **0.1821** | **0.0389** | **0.6462** | **0.5253** | **0.0690** | **0.2677** |
| | | U2U-R | 0.6931 | 0.2646 | 0.0370 | 0.7193 | 0.0704 | 0.0985 | 0.7107 | 0.5650 | 0.3536 | 0.4088 |
| | | D2D-R | 0.6720 | 0.2434 | **0.0529** | 0.6987 | 0.2359 | 0.0298 | 0.6728 | 0.5480 | 0.3103 | 0.2802 |
| **Random Attacker** | | | 0.5000 | 0.1429 | 0.0476 | 0.5000 | 0.1429 | 0.0476 | 0.5000 | 0.5000 | 0.1469 | 0.1667 |

**Table 4: Utility Results of Recommendation Performance.**

| Datasets | | Methods | Utility Metrics | | | |
|---|---|---|---|---|---|---|
| | | | NDCG@5 | NDCG@10 | HR@5 | HR@10 |
| **MovieLens-100K** | | Original | 0.7452 | 0.7080 | 0.7234 | 0.6804 |
| | DD | Aegis-Fed | **0.7632** | **0.6905** | **0.7321** | **0.6959** |
| | | UC-FedRec | 0.6959 | 0.6452 | 0.7032 | 0.6698 |
| | DF | Aegis-CS | **0.7209** | **0.6891** | 0.6944 | 0.6571 |
| | | U2U-R | 0.7151 | 0.6854 | 0.7040 | 0.6663 |
| | | D2D-R | 0.7194 | 0.6862 | **0.7129** | **0.6693** |
| **MovieLens-1M** | | Original | 0.6992 | 0.6901 | 0.6958 | 0.6843 |
| | DD | Aegis-Fed | **0.6332** | 0.6262 | **0.6292** | **0.6211** |
| | | UC-FedRec | 0.6320 | **0.6525** | 0.6280 | 0.6204 |
| | DF | Aegis-CS | 0.6631 | **0.6707** | **0.6641** | 0.6604 |
| | | U2U-R | 0.6481 | 0.6156 | 0.6420 | 0.5965 |
| | | D2D-R | **0.6928** | 0.6623 | 0.6388 | **0.6929** |
| **ModCloth** | | Original | 0.6077 | 0.6079 | 0.6047 | 0.6115 |
| | DD | Aegis-Fed | **0.6044** | **0.6071** | **0.6352** | **0.6386** |
| | | UC-FedRec | 0.5664 | 0.5551 | 0.5622 | 0.5542 |
| | DF | Aegis-CS | **0.5969** | **0.5814** | **0.5563** | **0.5530** |
| | | U2U-R | 0.5605 | 0.5437 | 0.5482 | 0.5319 |
| | | D2D-R | 0.5854 | 0.5694 | 0.5653 | 0.5561 |
| **Last.FM-1K** | | Original | 0.5724 | 0.5665 | 0.5806 | 0.5680 |
| | DD | Aegis-Fed | **0.5888** | **0.5939** | **0.5962** | **0.5999** |
| | | UC-FedRec | 0.5182 | 0.5190 | 0.5243 | 0.5229 |
| | DF | Aegis-CS | **0.5446** | 0.5726 | **0.5442** | 0.5052 |
| | | U2U-R | 0.5182 | 0.5190 | 0.5244 | 0.5229 |
| | | D2D-R | 0.5282 | **0.5975** | 0.5362 | **0.5490** |

Aegis-Fed performs better than UC-FedRec, reducing attack performance by 14.26% and 3.99%, respectively. This is because UC-FedRec's reliance on the MLP-based attribute filter limits its defense against XGBoost attackers. Moreover, Aegis, as a post-training privacy protection method, offers more flexibility without the need to modify the model modules. Third, in the data-free DF centralized protection scenario, Aegis-CS outperforms U2U-R and D2D-R. For MLP attackers, Aegis-CS reduces attack performance by 14.43%, compared to 7.22% and 10.30% reductions for U2U-R and D2D-R, respectively. For XGBoost attackers, U2U-R fails to deceive attackers and instead significantly increases attack performance, with an average increase of 90.99%. In contrast, Aegis-CS and D2D-R reduce attack performance by 12.94% and 5.96%, respectively.

### 4.2.2 *Recommendation Performance*.
The recommendation performance evaluation based on Normalized Discounted Cumulative Gain (NDCG) and Hit Rate (HR) is shown in Table 4. We find that attribute unlearning methods can impact recommendation performance to varying degrees. Specifically, in the data-free DF centralized protection scenario, Aegis-CS results in an average reduction of 2.48% and 1.48% for NDCG@5 and NDCG@10, and a 3.63% and 4.21% decrease in HR@5 and HR@10, respectively. U2U-R exhibits larger declines across these four metrics, averaging 4.57%, 5.22%, 4.65%, and 5.67%. In contrast, D2D-R shows comparable performance to Aegis-CS, with average decreases of 2.47%, 1.43%, 3.78%, and 1.92%, respectively, outperforming U2U-R. In the DD scenario, Aegis-Fed achieves better recommendation performance than Aegis-CS, with average declines of only 0.87%, 1.37%, 0.30%, and 0.28% across the four metrics, and even improves performance in some instances. This improvement might be attributed to the reduction of attribute bias, leading to a more balanced data distribution. In comparison, UC-FedRec shows over a 4% average decrease across all recommendation performance metrics.

### 4.2.3 *Visualization*.
To visually analyze the results, we employ t-SNE [56] method to reduce the dimensionality of user embeddings to 2 for display. Figures 2a and 2b illustrate the gender distribution of users in the MovieLens-100K dataset before and after attribute unlearning, respectively, while Figures 2c and 2d present the occupational distribution of users in the MovieLens-1M dataset before and after unlearning. Since the goal of attribute unlearning is to make different categories of private attributes indistinguishable, the visualizations reveal a more diffused and less distinct distribution of privacy categories after unlearning, which reduces the likelihood of user embeddings being identifiable by attackers.

### 4.2.4 *Efficiency*.
We recorded the runtime of the unlearning methods for the gender attribute on the MovieLens-100K, MovieLens-1M, and Last.FM-1M datasets, as well as the body shape attribute on the ModCloth dataset, to reflect the different efficiency. The results are presented in Table 5. We observe that in the data-dependent DD scenario, Aegis demonstrates superior efficiency compared to the

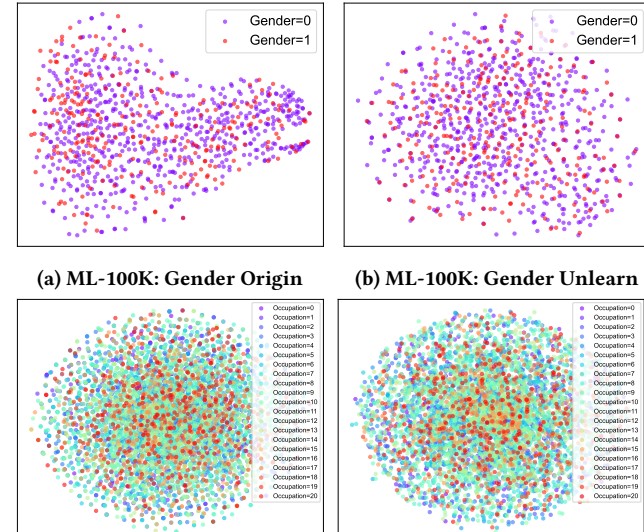

(a) ML-100K: Gender Origin   (b) ML-100K: Gender Unlearn

(c) ML-1M: Occupation Origin   (d) ML-1M: Occupation Unlearn

**Figure 2: t-SNE Visualization of User Embeddings.**

**Table 5: Running Time Consumption of Unlearning Methods.**

| Time(s) | Aegis-Fed | UC-FedRec | Aegis-CS | U2U-R | D2D-R |
|---|---|---|---|---|---|
| ML-100K | 392.04 | 941.74 | 10.91 | 9.80 | 5.43 |
| ML-1M | 4742.96 | 8223.53 | 37.66 | 109.68 | 54.45 |
| ModCloth | 571.22 | 981.43 | 165.18 | 167.37 | 87.35 |
| Last.FM-1K | 65085.79 | 163129.51 | 8.88 | 12.15 | 5.68 |

UC-FedRec. This efficiency advantage arises from Aegis's ability to perform federated fine-tuning on the existing trained global model without introducing additional MLP-based attribute filtering modules. In the data-free DF scenario, Aegis-CS significantly accelerates the unlearning process by eliminating the need for interactive data, achieving comparable efficiency to U2U-R and D2D-R. Our methods provide new insights into rapid attribute unlearning.

*4.2.5 **Ablation Study**.* To evaluate the robustness of our proposed multi-component loss function, we analyzed the effects of the trade-off parameters $\beta$ and $\gamma$. By fixing one parameter at its default setting and varying the other, we measured recommendation performance using HR@10 and unlearning effectiveness using attack accuracy, as shown in Figure 3. The results reveal that increasing $\beta$ leads to a slight improvement in recommendation performance and a minor decrease in unlearning effectiveness for both Aegis-Fed and Aegis-CS. In contrast, the sensitivity analysis of $\gamma$ shows the opposite effect. Overall, both Aegis-Fed and Aegis-CS demonstrate robustness across different parameter settings, with minimal variation in recommendation and unlearning performance.

## 5 RELATED WORK

**Recommendation Unlearning.** Machine unlearning has recently emerged as a method to quickly remove the impact of specific data on trained models [64]. Exact unlearning, exemplified by the SISA method [6], uses dataset partitioning and sub-model aggregation for fast retraining. Approximate unlearning manipulates model parameters, using techniques such as boundary learning [10] and knowledge distillation [35, 62]. Attribute unlearning is a type of

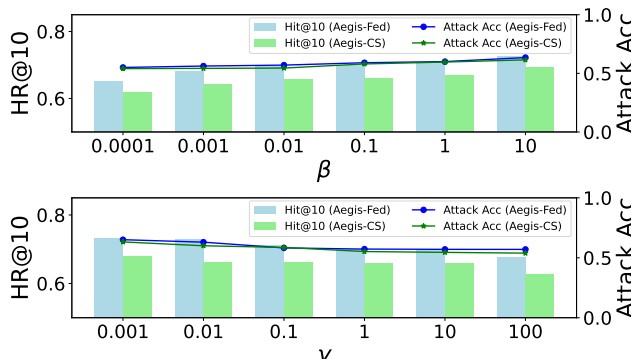

**Figure 3: Impact of $\beta$, $\gamma$ on Unlearning (Attack Accuracy) and Recommendation (HR@10) Performance on ML-100K.**

machine unlearning. Early research on attribute unlearning [22] tackled privacy in facial recognition by adding network modules. In recommender systems, methods like [67] adjust gradient updates, while others [9, 39] apply SISA-based data grouping for unlearning. However, most recommender system unlearning has focused on the sample level [37, 43], with limited work on federated attribute unlearning. While [40] explored attribute unlearning with a custom loss function, it hasn't been extended to federated settings. There's a need for a more robust federated attribute unlearning framework that works with or without access to training data.

**Privacy-preserving Recommender Systems.** Federated learning (FL) [46] has been applied in recommender systems to enhance user privacy by avoiding direct data sharing. However, federated recommender systems (FedRec) [47, 48] are still vulnerable to privacy risks, as central servers can infer private information from user parameters [55], such as user interactions, ratings, or attributes [8, 25, 61]. To mitigate these risks, privacy preservation mechanisms such as fake items, homomorphic encryption, and differential privacy (DP) have been employed. For example, [41, 42] use randomly sampled fake items to obscure user interactions, while homomorphic encryption [50, 66] ensures secure mathematical operations on encrypted data. Local differential privacy (LDP) [13, 15, 45] enables statistical computations and ensures the privacy of individual participants. However, encryption-based methods significantly increase communication costs, and DP primarily focuses on the transmission of model weights or updates, which is less effective in addressing inference attacks and privacy leakage in the learned federated recommender models [25, 55].

## 6 CONCLUSIONS

In this paper, we present Aegis, the first recommendation framework to address post-training attribute unlearning in a federated setting to our knowledge. Aegis enhances the privacy of FedRecs by selectively unlearning private user attributes while preserving recommendation quality. The framework balances private attribute unlearning and recommendation knowledge retention using a multi-component loss function based on information theory. Our approach minimizes the mutual information between user embeddings and sensitive attributes, combining this with regularization and recommendation losses to maintain performance. Aegis supports both interaction data-dependent and data-free unlearning, making it adaptable to different levels of data access.

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
