# OpenReview forum: "Aegis: Post-Training Attribute Unlearning in Federated Recommender Systems against Attribute Inference Attacks"
_ACM.org/TheWebConf/2025/Conference — WWW 2025 Poster_

### Official Review · Reviewer_H3vh · 2024-11-11

**Novelty:** 3
**Technical Quality:** 3

**Review:**

The manuscript presents a novel framework, Aegis, addressing post-training attribute unlearning in federated recommender systems to protect user privacy against attribute inference attacks. The paper is generally well-structured, with a clear abstract, introduction, methodology, and evaluation. However, there are several areas that require attention to improve the clarity and rigor of the presentation.

Strengths:

The paper has a clear structure, making it easy to follow the authors' line of thought.

Weaknesses:

Introduction: The logic in the introduction, particularly in paragraphs three and four, is not clear. The authors discuss current privacy protection methods and unlearning research, but their connection is not well-established, causing confusion for the reader.
The introduction emphasizes the dynamic and evolving nature of privacy protection. However, the subsequent model and experimental sections do not adequately address this aspect, leaving a gap between the problem statement and the solution.

Notation Consistency: The use of symbols in Equations 7 and 8 is inappropriate and could lead to misunderstanding. The authors should review these equations to ensure that the notation is consistent and clear.

Methodology Clarity: In Section 3.2, the authors propose two optimization objectives based on mutual information. However, the Private Attributes Information Loss section lacks a derivation process for the KL divergence upper bound. Additionally, the Recommendation Knowledge Retention Loss section does not mention how the mutual information objective is achieved, only adding a regularization term to the existing loss function.

Framework Generality: The authors claim that the Aegis framework is general and applicable to systems based on different training methods. However, this claim is not validated in the evaluation section, which limits the framework's versatility.

Visualization Clarity: Figure 2's visualization results do not show a significant difference before and after unlearning. The increased dispersion is not clearly demonstrated, and it is unclear whether the visualizations in (a) and (c) represent user embeddings or raw data. The authors should clarify this and ensure that the visualizations effectively highlight the importance of their work.

Complexity Analysis: The paper lacks a theoretical complexity analysis. Providing such an analysis would strengthen the manuscript by giving readers insight into the computational efficiency of the proposed framework.

**Questions:**

$\bullet$ Clarify the logic in the introduction, particularly about the current privacy protection methods.

$\bullet$ Address the dynamic and evolving nature of privacy protection in the model and experimental sections.

$\bullet$ Review and correct the notation in Equations 7 and 8 for clarity and consistency.

$\bullet$ Provide a detailed derivation for the KL divergence upper bound in the Private Attributes Information Loss section and explain how the mutual information objective is achieved in the Recommendation Knowledge Retention Loss section.

$\bullet$ Validate the generality of the Aegis framework with additional experiments or comparisons.

$\bullet$ Enhance Figure 2's visualizations to clearly show the differences before and after unlearning, and clarify what is being visualized.

$\bullet$ Include a theoretical complexity analysis to provide insight into the computational efficiency of the proposed framework.

**Reviewer Confidence:**

3: The reviewer is confident but not certain that the evaluation is correct

**Scope:**

4: The work is relevant to the Web and to the track, and is of broad interest to the community

---

### Official Review · Reviewer_SKiN · 2024-11-25

**Novelty:** 5
**Technical Quality:** 6

**Review:**

This paper focuses on the privacy risks associated with user attributes in federated recommendation systems and proposes a post-training unlearning method.

**Pros:** The research topic is valuable and practical; the method employs mutual information and Gaussian modeling, demonstrating technical improvement; and the experiments are sufficient in quantity.

**Cons.:**  My primary concerns lie in several hypothetical scenarios: Firstly, in existing research on FRS, user embeddings have gradually been discarded due to privacy concerns. In papers such as FedRAP and pFedRec, linear layers have replaced collaborative filtering algorithms for prediction, making FedNCF a relatively outdated approach. In this context, does Aegis still hold research value? Secondly, I do not believe that privacy requirements can easily change over time and users' privacy information tends to be relatively fixed. Moreover, why would users choose a post-training approach to protect their privacy? Information often changes over time, and the importance of privacy protection tends to decrease rather than increase. If privacy information can only be confirmed after training, how can we ensure that other clients do not retain models that previously contained sensitive information? Lastly, the authors also assume data deletion, which raises doubts. Whether local training data is accessible is a significant research issue, often arising in scenarios such as continual learning. The authors do not further analyze this aspect and simply assume that the data for unlearning is no longer accessible.

**Questions:**

Regarding the design of Aegis, I have no issues and consider the techniques to be advanced and innovative. I will ultimately adjust my judgment on the novelty based on the authors' explanation of the **Cons.**; otherwise, I can only consider that the authors have created an unrealistic and valueless task scenario.

**Minor points:** The authors may consider rephrasing the Problem Formulation to reduce redundancy in notation. For example, what does "M" mean in Line 306? The function naming convention like "dg" in Line 284 could also be improved. Additionally, there is a missing space in Line 107. The explanation of Table 1 is not clear enough, which may prevent readers from immediately understanding the significance of these experimental results. The authors should provide a clearer interpretation of the table to help readers comprehend the results.

**Reviewer Confidence:**

4: The reviewer is certain that the evaluation is correct and very familiar with the relevant literature

**Scope:**

4: The work is relevant to the Web and to the track, and is of broad interest to the community

---

### Official Review · Reviewer_vJDK · 2024-11-28

**Novelty:** 5
**Technical Quality:** 5

**Review:**

This paper presents Aegis, a post-training federated unlearning framework for recommender systems to protect user privacy by removing sensitive attributes from embeddings. Aegis uses an information-theoretic loss function to balance privacy and recommendation quality, even without training data. Evaluations show it effectively safeguards privacy while maintaining high-quality recommendations.

Pros:
- Attribute information leakage is a significant problem in data privacy, which still cannot be effectively solved even in the federated learning scenario.
- The strategy designed based on mutual information is ingenious from the perspective of balancing privacy and recommendation accuracy. Methodologies are detailed and well-explained.
- The experiments are detailed, though more analysis and discussions are needed to help the readers better understand some illustrations. Besides, experiments should also be conducted based on non-IID settings for federated learning consideration.

Cons:
- The strategy proposed in this paper does not reflect proper compatability with the federated learning scenario. The following heterogeneity issues need to be considered and discussed.
   - The threat model seems to be designed based on IID data distributions. If not, how to ensure that the shadow dataset has the same distribution as the original dataset?
  - Based on Eq. (15), the privacy attributes information loss is designed based on a uniform prior distribution. Is it a proper setting if clients have non-IID data distributions? How about defining client-level prior distributions, i.e. each client having a different prior distribution based on its data distribution? Personalization has been considered in several existing works to balance local and global features in federated learning. It seems that client-level prior distributions can still ensure attribute privacy, and recommendation quality may get further improved.
  - In Section I, the authors mention that federated clients may want to adjust their privacy settings. I am wondering if this can further cause privacy heterogeneity, i.e. difference in $\beta$ and $\gamma$ across clients. If so, how will this situation be tackled?
  - The experiments should be conducted based on non-IID data distributions. If it is already considered, the relevant experimental settings need to be detailed.
- Fig. 2 needs further explanations. I cannot figure out significant difference between subfigures before and after attribute unlearning, especially for the occupational distribution.

**Questions:**

- In Eq. (9), the prior distribution is denoted by a conditional probability, which is not a standard description.
- There are some typos:
  - Right column in page 1, "... as attacks may come from complex machine learning or deep learningmodels [44], ...", "learningmodels" should be corrected to "learning models".
  - Right column in page 6. "..., but most of they are not applicable to the attribute unlearning problem.", "most of they" should be corrected to "most of them".

**Reviewer Confidence:**

3: The reviewer is confident but not certain that the evaluation is correct

**Scope:**

4: The work is relevant to the Web and to the track, and is of broad interest to the community

---

### Official Review · Reviewer_UVkF · 2024-11-30

**Novelty:** 4
**Technical Quality:** 4

**Review:**

This paper studies attribute inference attacks on recommender systems under the federated learning context. Specifically, a new defense based on machine unlearning is proposed to mitigate the attribute information leakage of the client to the server. Extensive experiments validate the effectiveness of the proposed method.

**Questions:**

1, The Threat model is unclear to me. The authors argue that “In this scenario, we assume the attacker adopts a grey-box attack strategy.” However, in an FL scenario, the server has full access to the local model. Why is a grey-box assumption assumed here? In addition, the authors form the attribute inference attack problem as a classification problem. Why does the server know all attributes in advance? In an FL environment, the server is expected to know nothing about the data.

2, The assumption of a shadow dataset does not make too much sense to me. Again, as a server, if a server in FL knows the shadow dataset, it almost means the server knows everything. This violates the assumption and motivation of federated learning.

3, Unlearning setting. Who and when implements unlearning? From this paper, it seems the client will take the role of implementing unlearning after the model is trained. There are several concerns in this setting. First, during training, the server can store the historical model updates to implement attribute inference attacks. This means attribute privacy is already leaked. Second, if protecting attribute information should be implemented during training, why not directly implement training-based attribute defense during the training phase? For me, the unlearning process is either not reasonable or necessary.

4, I find this paper is very similar to the paper [RA] as follows, while a comparison is missing.

[RA] Chen, C., Zhang, Y., Li, Y., Meng, D., Wang, J., Zheng, X., & Yin, J. (2024). Post-Training Attribute Unlearning in Recommender Systems. arXiv e-prints, arXiv-2403.

**Reviewer Confidence:**

4: The reviewer is certain that the evaluation is correct and very familiar with the relevant literature

**Scope:**

3: The work is somewhat relevant to the Web and to the track, and is of narrow interest to a sub-community

---

### Official Review · Reviewer_Dw4G · 2024-12-03

**Novelty:** 5
**Technical Quality:** 5

**Review:**

In summary, this paper proposes a federated approach to unlearning private attributes, leveraging both data interaction and data-free methods. The data interaction approach involves client participation in the unlearning process, while the data-free method does not require such interaction. The key objectives of the paper are to effectively unlearn private attributes while maintaining the model's performance. To evaluate the unlearning process, the paper employs Attribute Inference Attacks and uses FedRec as the underlying model. The paper is well-written and clearly outlines its contributions. However, it considers a limited range of private attributes, and it remains unclear how the quantity and quality of private attribute measurements influence the unlearning process and overall model performance.

**Questions:**

Line 385: How do you identify private attributes? What criteria determine whether an attribute is considered private?
How do you measure the contribution of attributes, particularly private ones, to the model's performance?
Line 419: What is the computational overhead of determining the distribution for each class at the client side? What assumptions are made about the clients' resource capacities?
Line 485: Does "post-training" imply that the model has already converged? If so, what role does the loss function play in the unlearning process? Is it correct to assume that no further training is required?
Line 524: What does the input 𝜓0 represent?
Line 528: What assumptions are made regarding the ratio of private attributes to the total number of attributes? How does this ratio impact the unlearning process?
Line 587: What about the global model? Does it have knowledge of the attributes that are unlearned, or is this information confined to the client’s local model?
Line 710, Table 3: The paper shows that UC-FedRec outperforms Aegis for the MLP Attack (DD) at least once across three datasets. What accounts for this performance difference?
Line 727: Similar to the comment on Line 710, why does Aegis underperform in HR cases, particularly in the MovieLens and Last.FM datasets?
Line 835, Table 5: It is unclear what metric is being used in the table—is it seconds or minutes? How does this impact client resource utilization?

**Reviewer Confidence:**

1: The reviewer's evaluation is an educated guess

**Scope:**

3: The work is somewhat relevant to the Web and to the track, and is of narrow interest to a sub-community